# Managing Persistent Subsolid Nodules in Lung Cancer: Education, Decision Making, and Impact of Interval Growth Patterns

**DOI:** 10.3390/diagnostics13162674

**Published:** 2023-08-14

**Authors:** Yung-Chi Liu, Chia-Hao Liang, Yun-Ju Wu, Chi-Shen Chen, En-Kuei Tang, Fu-Zong Wu

**Affiliations:** 1Department of Radiology, Xiamen Chang Gung Hospital, Xiamen 361028, China; gigiliu1974@hotmail.com; 2Department of Imaging Technology Division, Xiamen Chang Gung Hospital, Xiamen 361028, China; 3Department of Healthcare Administration Department, Xiamen Chang Gung Hospital, Xiamen 361028, China; 4Department of Biomedical Imaging and Radiological Sciences, National Yang-Ming University, Taipei 112304, Taiwan; herrick@herricane-med.com.tw; 5Department of Radiology, Kaohsiung Veterans General Hospital, Kaohsiung 81362, Taiwan; yjwu@vghks.gov.tw; 6Department of Software Engineering and Management, National Kaohsiung Normal University, Kaohsiung 80201, Taiwan; 7Physical Examination Center, Kaohsiung Veterans General Hospital, Kaohsiung 81362, Taiwan; chirise@vghks.gov.tw; 8Department of Surgery, Kaohsiung Veterans General Hospital, Kaohsiung 813414, Taiwan; ektang@vghks.gov.tw; 9School of Medicine, College of Medicine, National Sun Yat-Sen University, Kaohsiung 80424, Taiwan; 10Faculty of Medicine, School of Medicine, National Yang Ming Chiao Tung University, Taipei 112, Taiwan; 11Institute of Education, National Sun Yat-Sen University, Kaohsiung 804241, Taiwan

**Keywords:** subsolid nodules, interval growth, overdiagnosis, lung cancer

## Abstract

With the popularization of lung cancer screening, many persistent subsolid nodules (SSNs) have been identified clinically, especially in Asian non-smokers. However, many studies have found that SSNs exhibit heterogeneous growth trends during long-term follow ups. This article adopted a narrative approach to extensively review the available literature on the topic to explore the definitions, rationale, and clinical application of different interval growths of subsolid pulmonary nodule management and follow-up strategies. The development of SSN growth thresholds with different growth patterns could support clinical decision making with follow-up guidelines to reduce over- and delayed diagnoses. In conclusion, using different SSN growth thresholds could optimize the follow-up management and clinical decision making of SSNs in lung cancer screening programs. This could further reduce the lung cancer mortality rate and potential harm from overdiagnosis and over management.

## 1. Introduction

With the popularization and application of low-dose computed tomography (LDCT) screening for lung cancer worldwide, a high prevalence of early lung adenocarcinoma spectra manifesting as subsolid nodules (SSNs) have been identified, especially in Asian countries [1,2,3,4,5]. The American National Lung Screening Trial, a clinical randomized trial, has verified that compared to the chest radiograph screening group, an LDCT lung screening in high-risk smoking groups can reduce the lung cancer mortality by 20%, and improve the all-cause mortality by 6.7% [6]. A recent systematic review and meta-analysis has demonstrated that the current evidence supports a significant reduction in lung cancer-related mortality with the utilization of LDCT for lung cancer screening in high-risk populations with heavy smoking exposure [7]. However, there is limited evidence regarding the optimal screening frequency and interval period. Randomized trials are still ongoing for LDCT lung cancer screening in non-smokers. However, several cohort studies have observed that the implementation of lung cancer screening in Asian non-smoking groups can gradually reduce lung cancer mortality with each passing year [8,9]. Unfortunately, relevant studies have also found that with an increase in the volume of LDCT lung cancer screening, the rate of overdiagnosis will also increase [3,10,11]. Most lung cancers in Asian non-smokers are lung adenocarcinoma spectrum lesions manifesting as SSNs. Therefore, an efficient evaluation of these SSNs with heterogeneous growth trends will become a major clinical issue in lung cancer screening programs for Asian non-smoking populations [12]. SSNs can be categorized as pure ground-glass nodules (pGGNs) or part-solid nodules (PSNs) according to the guidelines of the Fleischner Society for the management of SSNs [13,14]. Furthermore, in the context of a lung cancer screening setting, Lung-RADS offers appropriate guideline for managing SSNs [15]. According to previous literature reviews, as high as 90% of persistent pulmonary SSNs are early lung adenocarcinoma spectrum lesions after 3–6 months of follow up [16,17]. In the context of the subsolid nodules definition, persistent SSNs generally mean that there is no significant resolution or disappearance of the nodule observed within a three-month follow-up period [18]. Persistent SSNs presenting as lung cancer are often lung adenocarcinoma spectrum lesions, including atypical adenomatous hyperplasia, adenocarcinoma in situ, minimally invasive or invasive adenocarcinomas according to the histopathologic classification of adenocarcinoma of the lung, as reported by the International Association for the Study of Lung Cancer [19]. According to the 2021 WHO Classification of Lung Tumors, atypical adenomatous hyperplasia and adenocarcinoma in situ lesions are classified as precursor glandular lesions [20].

Previous studies have demonstrated that SSNs ≤ 3 cm in size have heterogeneous growth patterns, although most of these nodules have an indolent growth pattern [21,22,23]. A recent systematic review and meta-analysis has addressed that the pooled growth rate was 22% for SSNs and 26% for pGGNs using the definitions of SSN growth as follows: 2 mm or more increase in the longest diameter [24]. Furthermore, several studies have found that nodules, which remained stable in size for 2 or more years, have a very low incidence of growth (approximately 5%) on further surveillance [25,26,27]. However, the identification of SSNs that may grow faster and affect the clinical prognosis has become an important clinical problem [28]. Furthermore, the excessive use of medical and surgical interventions for slow-growing early lung adenocarcinomas and associated GGNs can lead to overdiagnosis in cancer screening programs, resulting in an unnecessary strain on healthcare resources [10]. Therefore, this review discusses the clinical impact of the different interval growth rates of SSNs and how to utilize this assessment for personalized pulmonary nodule follow-up and management recommendations. The definitions, study rationales, and clinical applications of the three different nodular growth patterns are discussed in the literature review, as shown in Figure 1.

## 2. Relevant Sections

### 2.1. SSNs Interval Growth with an Increase of ≥2 mm

Many studies have investigated the natural growth course and relevant risk factors of SSNs based on the definition of growth of ≥2 mm in lung nodules by their longest diameter [21,22,25,26,29,30,31,32,33,34,35,36]. According to previous studies addressing the interval change in SSN measurement of <2 mm subjected to measurement bias, an increase of ≥2 mm in size was considered significant in the definition of SSN interval growth (Figure 2) [26,37]. In addition, different acquisition parameters and kernel reconstruction could introduce measurement bias in assessing nodule interval growth [38]. However, the nodule growth definition based on the 2 mm growth threshold does not allow further assessment of trends in the natural growth course of SSNs that may affect clinical outcomes or survival. The clinical management of pulmonary SSNs in lung cancer screening is often a dilemma in clinical decision making. While Lung-RADS guidelines offer clear recommendations for tracking and managing subsolid nodules, real-world scenarios may involve delays in diagnosis due to patient anxiety and concerns regarding nodule growth. Pulmonologists may also have concerns about potential medical litigation resulting from delays in diagnosis. On one end of the scale, the over management of small pGGNs (usually defined as a nodular diameter of less than 10 mm) meeting the 2 mm growth threshold may lead to increased overdiagnosis. The other end of the scale uses a follow-up strategy to monitor the subsolid pulmonary nodules. The delay in diagnosis and its impact on survival prognosis have considerable clinical implications for patients. According to a recent systematic review and meta-analysis clarifying the risk factors for SSN growth based on the definition of a nodule growing > 2 mm in diameter, male sex, history of lung cancer, a nodule size > 10 mm, nodule consistency, and age > 65 years were independent risk factors for SSN growth during the follow-up period of 24.2–112 months [39]. For pGGNs growth prediction, eight clinical or radiologic features, including male sex, smoking history, nodule size > 10 mm, larger nodule size, air bronchogram, higher mean CT attenuation, well-defined border, and lobulated margin, were independent risk factors. In the era of widespread lung cancer screening and to prevent overdiagnosis or delays in diagnosis and management, the implementation of risk-based follow-up guidelines for subsolid nodules (SSNs) or ground-glass nodules (GGNs) can assist in clinical decision-making and follow-up strategies. However, future research is needed to clarify the prediction of the growth threshold that affects the clinical prognosis or morbidity in patients with SSNs.

### 2.2. SSNs Interval Growth with an Increase of ≥5 mm

According to a previous literature review, some studies have defined nodule growth based on a growth threshold of ≥5 mm [21]. The main research rationale was that the growth threshold of 5 mm was the obvious or substantial SSNs interval growth during the follow-up period of CT scans through a visual assessment by radiologists or clinical physicians (Figure 3). Based on the nodule growth threshold of 5 mm, clinicians and radiologists can easily evaluate and compare the growth changes of nodules visually before and after the CT; defining an interval growth threshold of 5 mm is clinically feasible for clinical practice in the real-world setting. Tang et al. investigated the natural course of SSN in terms of substantial SSN growth based on the definition of an obvious increase of ≥5 mm in SSNs or solid portion in PSN from the baseline CT scan to detect substantial interval growth during interval CT scans [21]. Tang et al. demonstrated that the PSN group had a significantly higher growth rate than the GGN group in terms of substantial SSN growth. PSN group had an estimated 67.3% growth rate during the 5-year follow up. GGN group had an estimated 10.6% growth rate during the follow up.

The mean growth time for GGNs to reach a growth threshold of 5 mm was 9.426 years, whereas the PSN group had a mean growth time of 3.960 years to reach the threshold. Compared with a growth threshold of 2 mm, a growth threshold of 5 mm is more suitable for time-series clinical or radiologic evaluation of interval growth in SSNs. For example, one 3 mm GGN grows to 6 mm after 6 years, and such a GGN lesion meets the growth threshold of only 2 mm. A 6 mm GGN is very likely to be a pre-cancerous lesion or carcinoma in situ according to the pathological classification of the lung adenocarcinoma spectrum (Figure 4) [40]. However, if we just reach the growth threshold of 2 mm to make clinical decisions with surgical intervention, it may lead to overdiagnosis. Therefore, the use of a growth threshold of 5 mm for evaluation will increase the feasibility of clinical decision making and management of SSNs and reduce overtreatment and overdiagnosis in real-world lung cancer screening programs [41,42]. However, few studies have used this growth threshold (5 mm) to evaluate the SSNs growth rate. In the future, larger studies are needed with longer follow ups to verify the actual clinical benefit of the growth threshold of 5 mm in clinical nodule management and decision making.

### 2.3. SSNs Interval Growth with Clinical Stage Shift

Stage-shift growth in this study refers to the change in the prognostic stage based on the 8th lung cancer TNM classification system described in the previous relevant studies, assessed through CT scans [43]. It primarily reflects the transition between different stages (e.g., from II to III or 0 to I) and is influenced by the AJCC 8th edition of TNM staging, which considers the impact of interval growth progression on lung cancer survival prognosis. The clinical stage shift as the nodule growth threshold can be used as an important clinically personalized lung cancer prognostic indicator. Clinically, if the interval growth of nodules reaches this threshold, it may lead to a worse prognosis in patients with SSNs. The study rationale for this research lies in using the growth threshold of stage shift to identify the interval growth of SSNs and its correlation with prognostic clinical outcomes, as shown in Figure 5. Therefore, surgical interventions or clinical treatment strategies should be implemented promptly to avoid delayed diagnosis and management. Several studies have investigated the natural course of SSNs based on the definition of stage-shift interval growth from a baseline CT scan to detect stage-shift growth during interval CT scans [21,27]. Tang et al. demonstrated that the PSN group had a significantly higher growth rate than the GGN group in terms of stage-shift growth. PSN group had an estimated 24.2% growth rate within the 5-year follow-up period. GGN group had an estimated 0% growth rate during the follow up [21].

The mean growth time for GGNs to reach the growth threshold of the stage shift was 12.168 years, whereas the PSN group had a mean growth time of 7.198 years to reach the threshold point. Based on the interval growth threshold of stage shift in SSNs, clinicians and radiologists can identify high-risk SSNs based on initial CT parameters or clinical risk factors. By assessing the CT imaging characteristics and clinical profiles, they can predict further prognostic growth of these nodules. This information helps in determining the appropriate management strategy for patients with SSNs in order to identify any potential progression to invasive lung cancer associated prognostic outcome or stage-shift change. We also observed that certain clinical scenarios are particularly prone to rapidly worsening prognoses in patients presented with atypical peri-fissure SSNs [44]. Since atypical peri-fissure SSNs may be regarded as atypical peri-fissure nodules clinically, it is considered clinically benign without close follow up. However, since the lesion is close to the pleura surface or inter-fissure space, it is more likely to spread or metastasize along the inter-fissure space clinically [45,46]. Therefore, atypical peri-fissure SSNs should be tracked or dealt with more aggressively. Previous studies also have pointed out that if the subsolid nodule is close to the pleural surface or between pulmonary fissures, early lung cancer manifesting as SSNs could invade the inter-fissure space to form lung or pleural metastases/seeding (Figure 6) [21]. Compared to the growth threshold of 2 mm or 5 mm in SSNs, the clinical or radiologic risk features related to the growth threshold of stage shift represent the indicators that affect the prognosis of early lung adenocarcinoma. However, due to the emerging evidence-based medicine in SSN growth patterns, we have a better understanding of the growth history of SSNs, and have also pointed out that SSNs can be managed through active surveillance of follow-up strategies [24]. Currently, owing to the relevant medical evidence in SSN growth, a nodule growth threshold of 2 mm is often used as the basis for clinical decision making [24,39,42]. Moreover, it may be unacceptable in medical ethical issues to result in a poor prognosis of lung cancer due to a delayed diagnosis. Therefore, most relevant studies exploring the natural course of SSNs affecting the prognosis of growth adopt a retrospective research design for analysis. In the future, a comparison of the impact of different SSN growth patterns on clinical prognosis needs to be carried out through a more standardized image quantification method so as not to affect the patient’s prognosis or survival outcome. Clinicians or radiologists can better understand the clinical or radiologic high-risk factors that may affect the prognosis of lung cancer to guide individualized surgical decision plans or active surveillance plans for SSN management [13].

### 2.4. SSNs Interval Growth with Volumetric Assessment

Previous studies have used volumetric measurements to assess the interval growth change in SSNs [47,48,49,50,51,52,53]. Generally, the nodular volume can be measured in two ways. One is to measure the longest nodular diameter to calculate the volume. The other method involves segmenting and delineating the nodules to calculate the volume [54].

The main research rationale was that the European position statement recommended lung cancer screening follow up based on semi-automatically measured volume or volume-doubling time assessment of solid nodules (Figure 7) [55]. Several studies have reported the interval growth of pulmonary SSNs as a growth threshold of 20–30% volumetric increase for growth evaluation [47,48,49,50,51,52]. Previous studies have demonstrated that diameter-based assessments may overestimate the actual growth, as compared to growth assessments based on volume-based measurements [54]. In addition, both human and phantom studies have demonstrated that manual diameter measurements for nodule assessment are susceptible to errors influenced by nodule margin characteristics [56]. These characteristics can lead to either overestimation or underestimation of the true nodule size when measured manually [57]. Therefore, the volumetric assessment of SSN growth is clinically more sensitive than the nodule diameter measurement [48,58]. Therefore, it will be possible to avoid the delay of the disease or the rapid growth of nodules that affect the prognosis. However, if a volumetric growth assessment is used for small SSNs or GGNs < 5 mm, attention must be given to the problem of overdiagnosis or overtreatment due to the relatively high sensitivity in SSN growth thresholds based on volumetric measurements. Additionally, there are still some difficulties in the clinical practice of volumetric SSN measurements. Recently, an artificial intelligence (AI) analysis software has been used to measure volumetric parameters in a fully automatic or semi-automatic manner. However, clinical workflow integration and personalized pulmonary SSNs follow-up management platforms still need to be integrated with picture archiving and communication systems (PACS) and AI software in the future to practice more efficiently in the real world.

### 2.5. SSNs Interval Growth with Radiomic Assessment

Previous studies have demonstrated that radiomic assessment has achieved a good diagnostic performance in SSN interval growth pattern [59,60,61,62]. Gao et al. showed that the combined clinical and radiomic model showed good performance with an area under the curve (AUC) of 0.801 in predicting the 2 mm growth threshold of GGNs [59]. Furthermore, Sun et al. demonstrated that the radiomics model outperformed conventional radiographic parameters in GGN interval growth prediction [60]. Limited studies have also demonstrated radiomic features associated with SSN nodules with a high growth rate or shorter volume doubling time (VDT), especially margin-related radiomic features with higher volume-based doubling times in lung adenocarcinoma lesions manifesting as SSNs [61,62].

However, there are some limitations in radiomic-based research on the growth rate of longitudinal follow up of pulmonary nodules, such as the inconsistency of scanning protocols or vendors, which will affect the stability or robustness of radiomic features in interval growth prediction [63,64]. Moreover, delta-radiomics is a promising quantitative method that can assess serial changes through longitudinal CT imaging analysis, which could provide a potential growth tendency to guide high-and low-risk SSNs management [65]. Delta-radiomic-based imaging biomarkers have recently emerged as promising non-invasive tools for allowing a comprehensive evaluation of the tumor habitant microenvironment and the temporal interval changes of spatial heterogeneity in lung adenocarcinoma spectrum lesions [66,67,68]. Currently, radiomics approaches for SSN interval growth are primarily limited to research settings and have not yet gained widespread adoption in clinical practice. In the future, a combination of radiomics analysis software and PACS reporting platforms can optimize the efficiency and accuracy of radiomics analysis workflow in longitudinal SSNs series follow up.

### 2.6. Summative Umbrella and Narrative Review Approach for SSN Growth

We conducted an umbrella review from inception until 31 January 2023, with the following “OR” or “AND” search strategies (key words lists: subsolid nodule; ground-glass nodule; part-solid nodule growth; natural course; natural history; computed tomography) to review the relevant systemic reviews/meta-analyses/narrative reviews to address the current evidence and future direction on this issue about natural growth of SSNs. We also manually searched the relevant reference lists of eligible articles, narrative reviews, and editorials in this research field. Among them, four published reviews are included in this summative umbrella/narrative review (a summary review of these studies is shown in Table 1).

## 3. Discussion

### 3.1. Overdiagnosis

The purpose of this article is to examine and establish the relationship between SSNs and various interval growth patterns in terms of three aspects: measurement bias, radiologic follow up, and clinical prognosis. Therefore, we know that using a threshold of 2 mm or 20–30% volume increase is a more sensitive way to evaluate the nodule growth, and it is clinically suitable for evaluating larger SSNs to avoid the delay in diagnosis. If a 2 mm growth threshold is used to track small GGNs, it may be overly sensitive and may overestimate the growth of nodules, resulting in overdiagnosis and overtreatment. Establishing an SSN different interval growth prediction model based on clinical, radiologic, and radiomic risk features could guide clinically active surveillance of these SSNs and determine the optimal surgical timing in personalized lung cancer screening programs. Overdiagnosis is an inevitable byproduct of lung cancer screening, especially in non-smoking Asian populations [69,70]. Owing to the widespread application of LDCT for lung cancer screening in Asian countries, the incidental discovery of persistent SSNs with a high prevalence rate in Asian non-smokers also presents an important dilemma in clinical management and decision making [10]. Screening high-risk populations for lung cancer is a crucial step towards efficient and effective screening, ensuring that resources are focused on those who are most likely to benefit from early detection and intervention [12].

There is a notable knowledge gap in the medical field and a clinical challenge in accurately assessing individual lung cancer risk stratification among non-smokers in Asia, particularly women. Currently, there is controversy surrounding the eligibility criteria for lung cancer screening in the non-smoker population. Therefore, the widespread acceptance of self-paid lung cancer screening examinations in Asian countries may lead to ineffective screening programs and overdiagnosis/treatment [8,11]. The original purpose of screening is to find the clinically significant high-risk populations at the top of the iceberg [69,71]. However, if screening is not performed for high-risk groups, more preclinical or indolent-growing early stage lung cancer lesions in the form of persistent SSNs can occur, similar to the numerous asymptomatic or indolent-growing lung cancer lesions or precursor lesions under the iceberg phenomenon (Figure 8). We can solve the dilemma encountered in lung cancer screening programs in Asian non-smoking populations only by assessing the growth trend of SSNs through the SSN growth prediction model that can affect the clinical prognosis. Heterogeneous growth behavior of these SSNs (from indolent to rapid growth) may lead to delayed diagnosis or overdiagnosis if standardized nodule follow up or over management is neglected at the pre-cancerous stage. Combining multi-omics models to identify high-risk SSNs on the iceberg for early surgical intervention and active monitoring (wait-and-see policy) for a large number of low-risk indolent SSNs under the iceberg could optimize the overall quality of lung cancer screening [41]. Therefore, using different growth thresholds to evaluate the growth pattern of natural SSNs may be more effective in distinguishing clinically high-risk SSN lesions from subclinical indolent or stable SSN lesions. It is believed that through the development of personalized prediction models for different growth trends in SSNs, the SSNs management with follow-up strategies could be optimized to maintain a balance between the pros and cons in lung cancer screening programs. Overdiagnosis is inevitable in the screening process, similar to the tip of the iceberg [69,71,72]. However, through the evaluation of individualized nodule growth risk factors, initial size of SSNs, different growth patterns of SSNs, multidisciplinary teamwork, and patient education with a shared decision-making plan to maintain the two ends of the balance effect in lung cancer screening, the advantages and disadvantages of lung cancer screening can be realized in real-world practice (Figure 9).

The iceberg phenomenon is common in lung cancer screening programs. Screening for high-risk groups may help to identify more clinically significant lung cancers. However, targeting low-risk groups may find many subclinical or indolent lung cancers under icebergs. The use of different growth trend assessments can optimize individual subsolid nodule tracking and treatment strategies to reduce the occurrence of overdiagnosis and delayed diagnosis. SSNs, subsolid nodules.

At present, relevant studies have shown that CT guided biopsy is a safe and effective method for determining preoperative diagnosis of SSNs of suspected lung adenocarcinoma [73]. In a recent study conducted by Kiranantawat et al., the biopsy procedure demonstrated a high technical success rate of 94.7% [74]. The study also reported a 100% positive predictive value (PPV) for malignancy, indicating accurate identification of cancerous nodules. Additionally, the study observed a low rate of complications associated with the biopsy procedure. However, many previous studies have shown that active surveillance of persistent SSNs can effectively monitor the interval growth and avoid problems such as overdiagnosis and over management caused by excessive surgical or biopsy procedure [75,76]. Therefore, we suggest that active surveillance of SSNs based on follow-up guidelines will be the most efficient strategies than tissue sampling through CT-guided biopsy. It is recommended to biopsy solid nodules (originating from SSNs) that have developed to 2–3 cm in diameter or SSNs that have progressed to include a solid component measuring 2–3 cm in diameter, which will have more clinical significance. Due to the clinical possibility of metastasizing to mediastinal lymph nodes or distal metastasis in this clinical setting, a complete preoperative evaluation including CT-guided biopsy and positron emission tomography can better understand the comprehensive lung cancer staging status of patients and to determine the benign and malignant lesions. CT-guided biopsy can be performed in specific situations, such as determining the benign or malignant lesions or further comprehensively evaluating lung cancer staging with pathologic invasiveness degree.

### 3.2. Succinct Summary According to Summative Umbrella and Narrative Review

In this summative umbrella/narrative review, two were systemic reviews and meta-analyses, one study addressed only the systemic review, and one narrative review addressed the issue of SSN growth from the perspectives of experts in this field [24,39,41,42]. In this comprehensive review, most of these articles discussed the potential risk factors for the interval growth of SSNs. Some articles discussed the prevalence and differences in SSN growth in terms of nodular subtype during the follow-up period, and the issue addressing the nodule growth rate of SSNs that have been stable after a long-term follow up of 5 years [39]. In a systematic review by Gao et al., the authors concluded that CT attenuation may be useful in predicting the risk of SSNs growth [42]. However, this review also mentioned that the nodular diameter had a limited role in predicting the SSN interval growth. Due to heterogeneity among articles and differences in study design, the systematic review by Gao et al. did not perform pooling data analysis for further meta-analysis evaluation [42]. A meta-analysis by Wu et al. found that the pooled incidence of SSN growth rate was 22% based on a meta-analysis of 2898 SSNs among 16 enrolled studies [39]. In the subgroup analysis in terms of GGN growth, the pooled incidence of SSN growth rate was 26%. Moreover, the pooled incidence of SSN growth rate was only 5% after 2–5 years of stability, according to the subgroup meta-analysis results. The authors concluded that a long-term follow up with active surveillance of these SSNs was essential for SSN management.

However, the initial SSN size was found to predict the nodule interval growth in this study by Wu et al. but was not found in another systemic review by Cao et al. [24,42]. Sources of variability in the main findings may arise from differences in the initial nodular size from the enrolled studies. In the future, the use of personalized multivariate parameters to predict nodule interval growth may be more clinically beneficial for the growth prediction in the management of SSNs with different growth behaviors. A recent meta-analysis and systemic review by Liang et al. demonstrated that male sex, history of lung cancer, nodule size > 10 mm, nodule consistency, and age > 65 years were independent risk factors for predicting SSN growth [39]. Previous studies also demonstrated that the follow-up duration was significantly associated with SSN interval growth [21,32]. However, the study participants enrolled in this meta-analysis were not based on homogeneous populations and the presence of heterogeneity in the study design, which could limit the study outcome with external validity. A recent narrative review by Zhang et al. addressed the current evidence regarding clinical information with CT radiological features to predict SSN growth [41]. The results of the literature review showed that the interval growth threshold and patterns of SSNs were not standardized among the studies, and the related literature were diverse from Asian countries. Therefore, we can infer that this is related to the higher prevalence of subsolid nodules in the Asian population. In the future, it may be necessary to further standardize the interval growth threshold in SSNs and compare more Eastern and Western data for verification. Furthermore, future research should focus on how to use multi-omics combined with longitudinal delta-radiomics data or combined with tumor microenvironment habitant analysis to determine the heterogeneous growth pattern of these SSNs in their natural long-term follow-up course.

### 3.3. Future Direction

From the results of the current umbrella literature review, we know that the clinical risk factors (such as smoking, age, or family history of lung cancer) and conventional imaging characteristics (initial nodule size and nodule type) may play an important role in predicting the growth rate of SSNs. However, from this meta-analysis, we also found a high heterogeneity in the research design among different enrolled studies, such as follow-up period, nodule type, and initial nodular size. The relevant meta-analysis results showed a high degree of heterogeneity. Future study design should be directed towards standardization of study design, such as follow-up time, consistency of initial nodule size, and nodule type, to reduce the research bias. In addition, through the consistency of research design and imaging scan parameters, radiomics can be used to quantitatively analyze the delta-radiomics of SSNs at different time points, and combined with clinical, genetic, and tumor microenvironment parameters to construct a robust multiomics-based prediction model for SSN interval growth in the field of precise medicine.

## 4. Conclusions

This literature review addressing the different interval growths in SSNs with its clinical application, comprehensively discusses the definition, design rationale, and current applications of different SSN growth patterns and potential limitations. Summative umbrella reviews have also demonstrated current evidence regarding the natural growth of SSNs based on clinical and conventional CT characteristics. The summative umbrella review in this article also discussed the future research directions on the growth of SSNs. In the future, personalized SSN management guidelines can be further optimized by integrating multi-omics and the impact of different growth patterns on the clinical prognosis. This literature review could allow clinicians and radiologists to understand how to optimize the common dilemma in the process of planning lung cancer screening through different personalized SSN growth trend prediction models to minimize over- and delayed diagnoses, and maximize screening benefits.

## Figures and Tables

**Figure 1 diagnostics-13-02674-f001:**
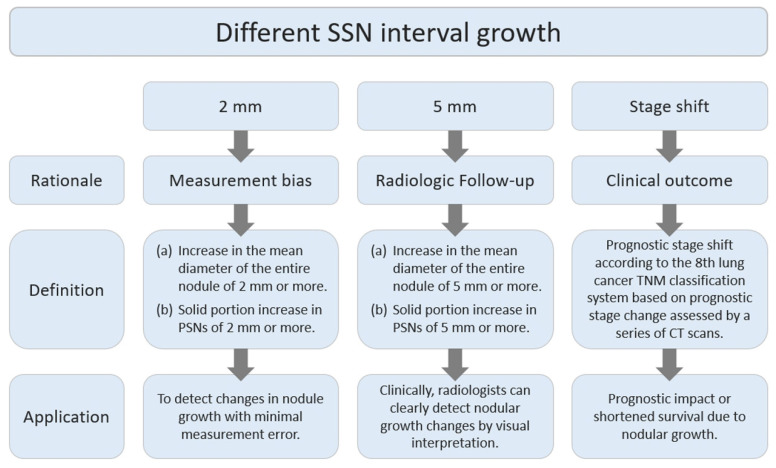
Rationale, definition, and clinical applications of different SSN interval growths in terms of 2 mm, 5 mm, and stage-shift interval growth. Definition: (a) Nodule’s mean diameter increases by ≥2 mm or ≥5 mm based on the threshold; (b) Solid portion in PSNs grows by ≥2 mm or ≥5 mm per the threshold. CT, computed tomography; PSN, part-solid nodule; and SSN, subsolid nodules.

**Figure 2 diagnostics-13-02674-f002:**
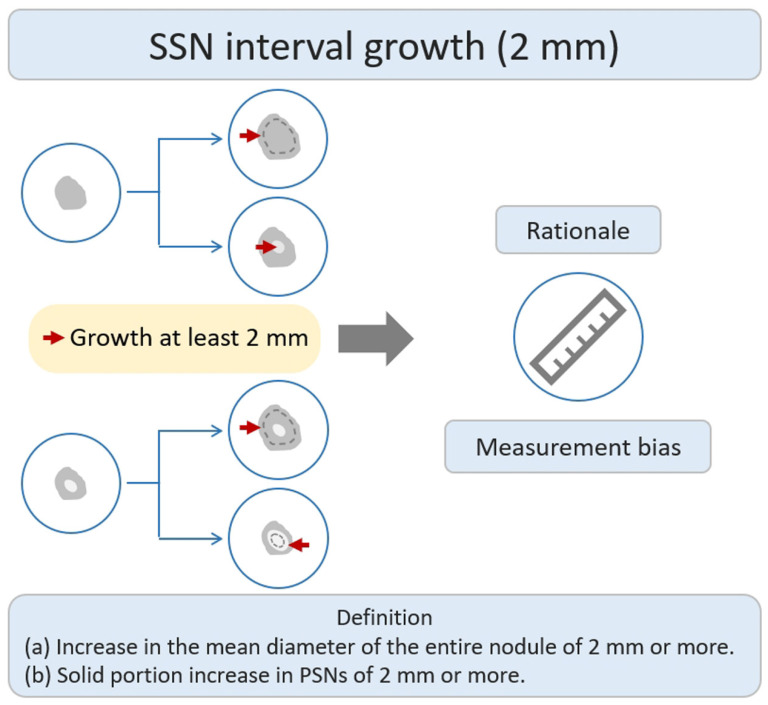
Definition, theoretical rationale, and clinical values based on interval growth threshold of 2 mm in SSN. The definition of growth (a & b) has been explained above. PSN, part-solid nodule; SSN, subsolid nodule.

**Figure 3 diagnostics-13-02674-f003:**
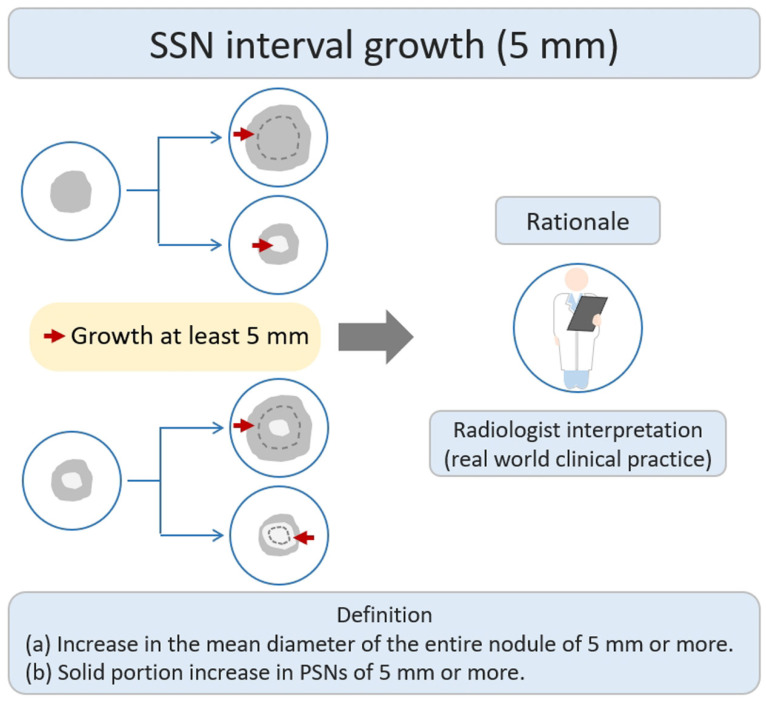
Definition, theoretical rationale, and clinical value based on interval growth threshold of 5 mm in SSN. The definition of growth (a & b) has been explained above. PSN, part-solid nodule; SSN, subsolid nodule.

**Figure 4 diagnostics-13-02674-f004:**
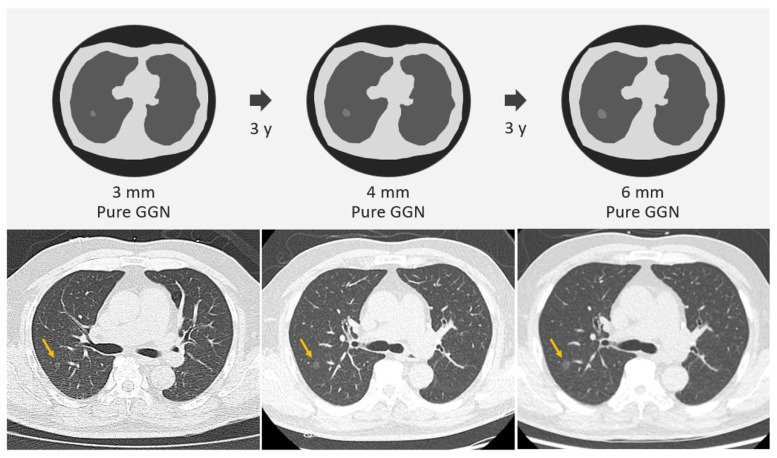
Example of one GGN 3 mm in a 50-year-old man in the right lower lobe. The first CT scan revealed a 3 mm GGN in the right upper lung (yellow arrows). After 3 years of follow up, the GGN had increased to 4 mm; however, it may be due to measurement errors with clinical follow-up strategies. After 3 years of tracking, the GGN had further increased to 6 mm, which meets the 2 mm threshold of interval growth. According to the pathological classification of lung adenocarcinoma spectrum, this lesion is most likely adenocarcinoma in situ. Therefore, if surgery is performed according to this growth threshold of 2 mm, the possibility of overdiagnosis and over management for indolent lung cancer will be greatly increased. Finally, the pathology report also confirmed the diagnosis of adenocarcinoma in situ. CT, computed tomography; GGN, ground-glass nodule.

**Figure 5 diagnostics-13-02674-f005:**
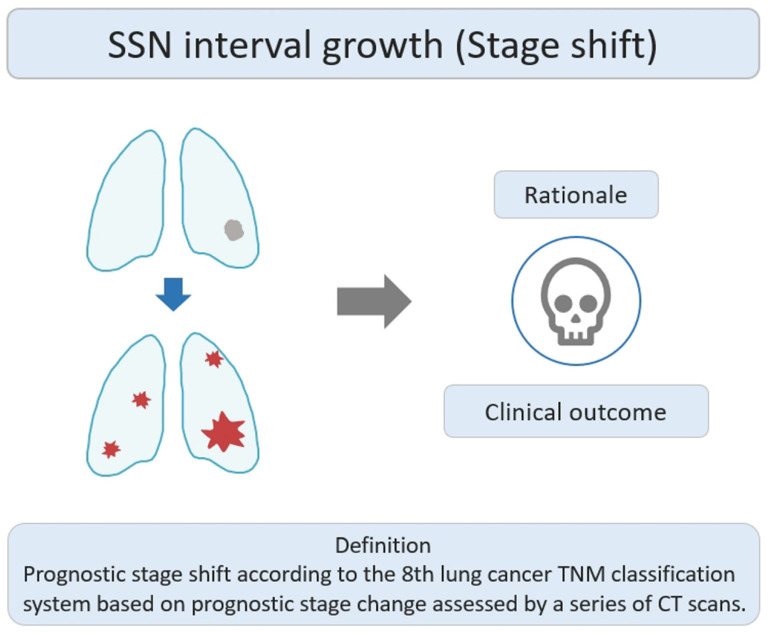
Definition, theoretical rationale, and clinical values based on interval growth threshold of stage shift in SSN. The visual illustration entails a single lung nodule’s evolution (gray color) into advanced lung cancer. Concurrently, numerous lung nodules (red color) emerge, indicating metastases and prompting a shift in the disease stage. CT, computed tomography; SSN, subsolid nodule.

**Figure 6 diagnostics-13-02674-f006:**
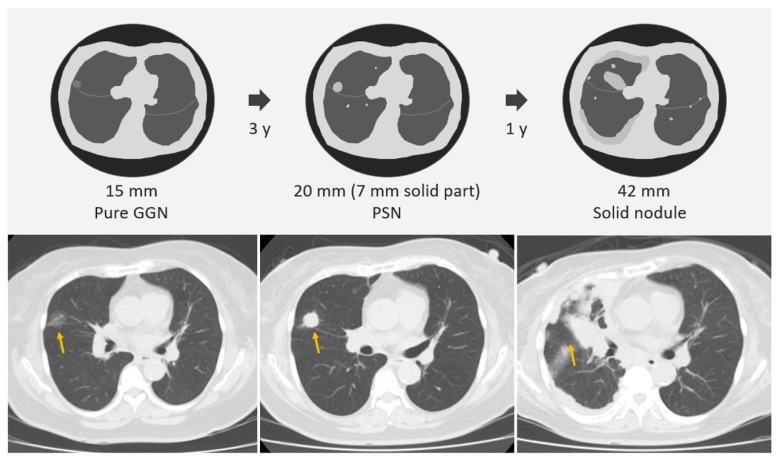
Example of one GGN 15 mm in a 65-year-old woman in the right middle lobe (yellow arrows). The first CT scan revealed a 15 mm GGN in the right upper lung. After 3 years of follow-up, the GGN had increased to 20 mm PSN with solid part measuring 7 mm. After 1 year of tracking, the PSN had increased to 42 mm (solid nodule) with inter-fissure invasion and pleural nodular seeding. Bilateral lung metastases also confirmed the stage shift in terms of SSN growth. Delayed diagnosis and treatment will seriously affect the survival and prognosis of such patients. Finally, the pathology report also confirmed the diagnosis of advanced invasive adenocarcinoma with lung and pleural metastases. CT, computed tomography; GGN, ground-glass nodule; PSN, part-solid nodule; and SSN, subsolid nodule.

**Figure 7 diagnostics-13-02674-f007:**
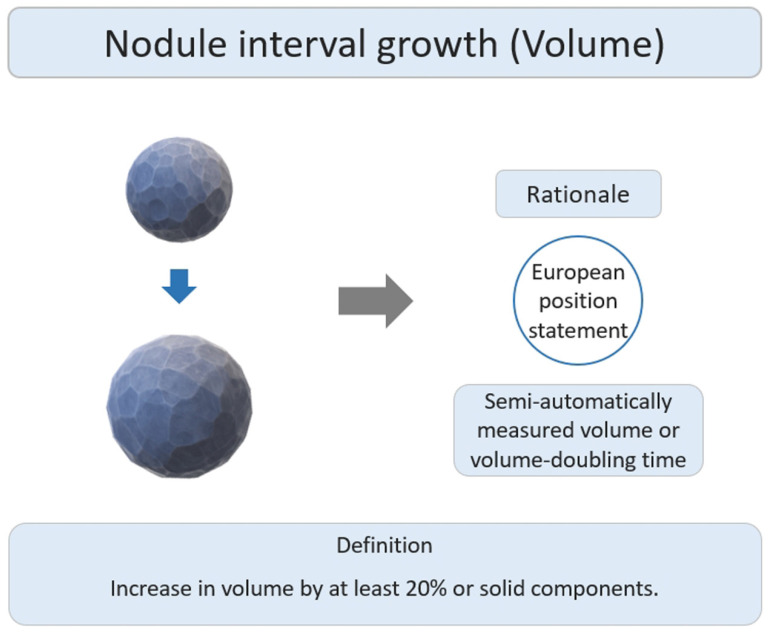
Definition, theoretical rationale, and clinical value based on interval growth threshold of volumetric assessment in subsolid nodules.

**Figure 8 diagnostics-13-02674-f008:**
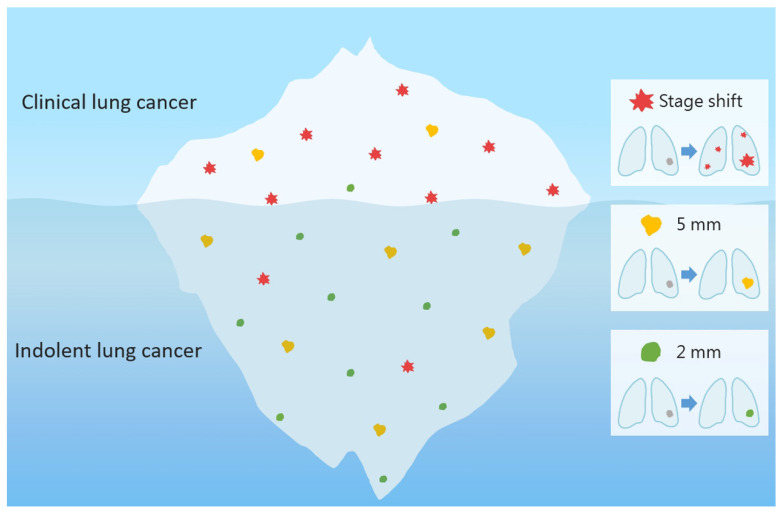
Iceberg phenomenon in interval growth of SSNs in lung cancer screening.

**Figure 9 diagnostics-13-02674-f009:**
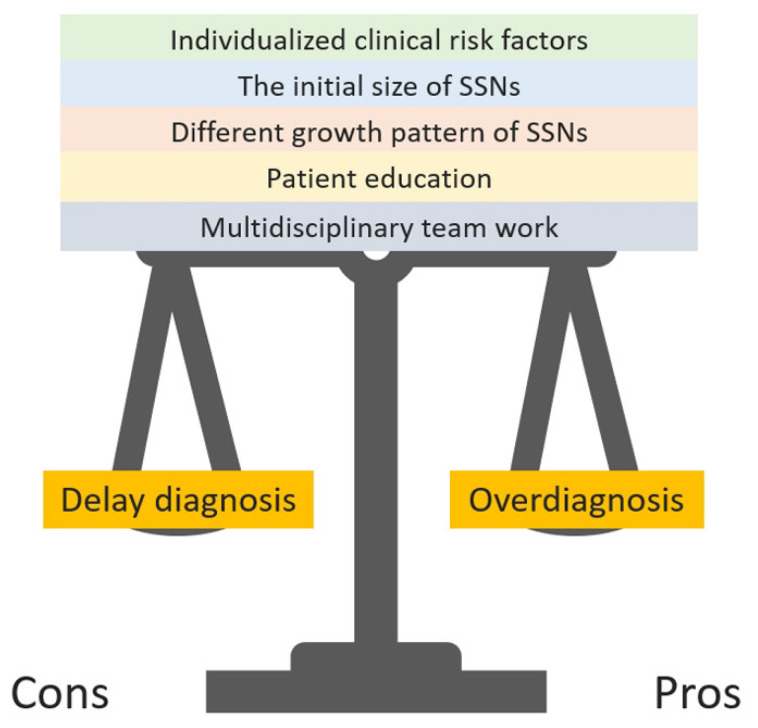
Five clinical strategies to help optimize common dilemmas in lung cancer screening: overdiagnosis and delayed diagnosis.

**Table 1 diagnostics-13-02674-t001:** Characteristics and key findings of the included systematic reviews/meta-analyses and narrative reviews (summative umbrella review).

1st Author, Year	Type	Enrolled Studies	Sample Size	Purpose	Main Findings	Limitations
Chen Gao, 2020 [59]	Systematic review	*n* = 10	850	Association between quantitative features of initial CT imaging and interval natural growth of SSNs to explore the potential risk factors.	CT attenuation in predicting the natural growth of SSNs.	1. Small sample size. 2. Regarding the variable inclusion criteria among studies. 3. Length of follow up varied.
Zhedong Zhang, 2022 [41]	Narrative review	N/A	N/A	Briefly describe and review the differential diagnosis, growth patterns and rates, genetic characteristics, and factors that influence the growth of persistent SSNs.	Predicting and quantitatively evaluating the growth of GGNs based on clinical and imaging feature data can provide a reference for the formulation of clinical diagnosis and treatment strategies for GGNs.	Liquid biopsy, multi-omics, and delta-radiomics prediction model development for further research direction.
Linyu Wu, 2022 [24]	Systematic review and meta-analysis	*n* = 16	2898 (available SSNs)	To estimate the incidence of interval growth after long-term follow up and identify the predictors of interval growth in SSNs on chest CT.	The pooled incidence of SSN growth was 22%, with a 26% incidence for pure GGNs.	The heterogeneity of SSNs in the included studies was high.
Xin Liang, 2022 [39]	Systematic review and meta-analysis	*n* = 19	2444 (3012 SSNs)	To identify clinical and CT risk features correlated with SSN interval growth.	Male sex, history of lung cancer, nodule size > 10 mm, nodule consistency, and age > 65 y were identified as independent risk factors for SSN growth.	1. The patients in the included studies were not completely homogenous. 2. Most of the features had no multivariate analysis.

CT, computed tomography; SSNs, subsolid nodules; GGNs, ground-glass nodules; and N/A: not applicable.

## Data Availability

The data are not publicly available.

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
