# Peer review of "Managing Persistent Subsolid Nodules in Lung Cancer: Education, Decision Making, and Impact of Interval Growth Patterns"

_diagnostics, 2023, doi:10.3390/diagnostics13162674_

Round 1

Reviewer 1 Report

This paper was a review and summary of the systematic reviews and current literature (definitions, rationale, and clinical applications) for interval growths of subsolid pulmonary nodule management.

Do not use the terminology "prediction modeling" (lines 23-24) since no analytic techniques were used to construct this review and this is misleading. You have created summary guidelines based on systematic reviews. Please remove all references to prediction modeling throughout the paper.

The figures throughout the paper are well constructed and helpful for clinical assessment, rationale, and review. I am not familiar with an umbrella review but I think an emphasis in the paper on how the systematic reviews/meta analysis were selected would give more utility and credibility to the manuscript. 

Table 1 needs to be cleaned up and removed from the discussion; no new results should be introduced in the discussion and a more succinct summary needs to be used in place of what is currently in the main finding section. 

I think this is a well written and organized manuscript and with a little additional work it will be a valuable paper.

Author Response

TITLE: RE:

Manuscript ID: Diagnostics-2520711

Title: Managing persistent subsolid nodules: education, decision-making, and impact of interval growth patterns

We are grateful to the Reviewers for their valuable comments and suggestions. We have now addressed all of the Reviewer’s comments, and provide a point-by-point response below. All changes are underlined in the text of the revised manuscript.

Reviewer 1:

This paper was a review and summary of the systematic reviews and current literature (definitions, rationale, and clinical applications) for interval growths of subsolid pulmonary nodule management.

Response

We appreciate your positive evaluation of the manuscript and your recognition of the importance of the research question we addressed. Your feedback regarding the clarity and organization of the manuscript was particularly valuable, and we have taken your suggestions into consideration during the revision process.

Major comment 1

Do not use the terminology "prediction modeling" (lines 23-24) since no analytic techniques were used to construct this review and this is misleading. You have created summary guidelines based on systematic reviews. Please remove all references to prediction modeling throughout the paper.

Response 1: We highly appreciate the reviewer’s comment. We have revised several sentences according to the reviewer’s comment in the abstract.

Major comment 2

The figures throughout the paper are well constructed and helpful for clinical assessment, rationale, and review. I am not familiar with an umbrella review but I think an emphasis in the paper on how the systematic reviews/meta analysis were selected would give more utility and credibility to the manuscript. 

Response 2: We highly appreciate the reviewer’s comment. We have revised several sentences according to the reviewer’s comment to address the process of the systematic reviews/meta-analysis/urembella review on the last paragraph in section 2.

Major comment 3

Table 1 needs to be cleaned up and removed from the discussion; no new results should be introduced in the discussion and a more succinct summary needs to be used in place of what is currently in the main finding section. 

Response 3: We highly appreciate the reviewer’s comment. We have revised the systemic review process to the section 2. And move the Table 1 to the section 2 according to the reviewer comment. Besides, we have revised several sentences according to the reviewer’s comment to address the succinct summary according to the literature review process in the discussion section.

Reviewer 2 Report

In this manuscript, Liu et al. reviewed the recent literature on persistent subsolid nodules in lung cancer. They discussed the definitions, rationale and clinical application of different interval growths of subsolid pulmonary nodule management. They analyzed the 2mm, 5mm and stage shift of different SSN interval growth. This review is good and suitable to publish in our journal. Because the subsolid nodules are mainly present in lung cancer, I suggest that the manuscript title revised to “Managing persistent subsolid nodules in lung cancer: education, decision-making, and impact of interval growth patterns.”

Author Response

TITLE: RE:

Manuscript ID: Diagnostics-2520711

Title: Managing persistent subsolid nodules: education, decision-making, and impact of interval growth patterns

We are grateful to the Reviewers for their valuable comments and suggestions. We have now addressed all of the Reviewer’s comments, and provide a point-by-point response below. All changes are underlined in the text of the revised manuscript.

Reviewer 2:

In this manuscript, Liu et al. reviewed the recent literature on persistent subsolid nodules in lung cancer. They discussed the definitions, rationale and clinical application of different interval growths of subsolid pulmonary nodule management. They analyzed the 2mm, 5mm and stage shift of different SSN interval growth. This review is good and suitable to publish in our journal. Because the subsolid nodules are mainly present in lung cancer, I suggest that the manuscript title revised to “Managing persistent subsolid nodules in lung cancer: education, decision-making, and impact of interval growth patterns.”

Response

We highly appreciate the reviewer’s comment. We have revised “the title “ according to the reviewer’s comment.